# Imaging of Peritoneal Carcinomatosis in Advanced Ovarian Cancer: CT, MRI, Radiomic Features and Resectability Criteria

**DOI:** 10.3390/cancers15245827

**Published:** 2023-12-13

**Authors:** Valentina Miceli, Marco Gennarini, Federica Tomao, Angelica Cupertino, Dario Lombardo, Innocenza Palaia, Federica Curti, Sandrine Riccardi, Roberta Ninkova, Francesca Maccioni, Paolo Ricci, Carlo Catalano, Stefania Maria Rita Rizzo, Lucia Manganaro

**Affiliations:** 1Department of Radiological, Oncology and Patological Sciences, “Sapienza” University of Rome, 00185 Rome, Italy; valentina.miceli@uniroma1.it (V.M.); marco.gennarini@uniroma1.it (M.G.); angelica.cupertino@uniroma1.it (A.C.); dario.lombardo@uniroma1.it (D.L.); federica.curti@uniroma1.it (F.C.); sandrine.riccardi@uniroma1.it (S.R.); robertavalerieva.ninkova@uniroma1.it (R.N.); francesca.maccioni@uniroma1.it (F.M.); paolo.ricci@uniroma1.it (P.R.); carlo.catalano@uniroma1.it (C.C.); 2Department of Gynecological, Obstetrical and Urological Sciences, “Sapienza” University of Rome, 00185 Rome, Italy; federica.tomao@uniroma1.it (F.T.); innocenza.palaia@uniroma1.it (I.P.); 3Clinica di Radiologia EOC, Istituto Imaging della Svizzera Italiana (IIMSI), 6900 Lugano, Switzerland; stefania.maria.rita.rizzo@usi.ch; 4Facoltà di Scienze Biomediche, Università della Svizzera Italiana (USI), 6900 Lugano, Switzerland

**Keywords:** ovarian cancer, peritoneal disease, radiomic, imaging, cytoreduction

## Abstract

**Simple Summary:**

Ovarian cancer is the second most frequent gynecological cancer in Western countries and the most common cause of death due to gynecological malignancies with an estimated five-year survival rate of 39%. The high aggressiveness and mortality are mainly related to the speed of abdominal spread: 70% of patients are diagnosed at an advanced stage of disease (stage III–IV FIGO) or in the presence of peritoneal carcinomatosis (PC), and about 60% of women will develop a recurrence. In this context, imaging plays an essential role for proper staging and follow-up and in selecting patients eligible for complete cytoreduction (CCR), the most important treatment and prognostic factor for patients.

**Abstract:**

PC represents the most striking picture of the loco-regional spread of ovarian cancer, configuring stage III. In the last few years, many papers have evaluated the role of imaging and therapeutic management in patients with ovarian cancer and PC. This paper summed up the literature on traditional approaches to the imaging of peritoneal carcinomatosis in advanced ovarian cancer, presenting classification systems, most frequent patterns, routes of spread and sites that are difficult to identify. The role of imaging in diagnosis was investigated, with particular attention to the reported sensitivity and specificity data—computed tomography (CT), magnetic resonance imaging (MRI), positron emission tomography-CT (PET-CT)—and to the peritoneal cancer index (PCI). In addition, we explored the therapeutic possibilities and radiomics applications that can impact management of patients with ovarian cancer. Careful staging is mandatory, and patient selection is one of the most important factors influencing complete cytoreduction (CCR) outcome: an accurate pre-operative imaging may allow selection of patients that may benefit most from primary cytoreductive surgery.

## 1. Introduction

Ovarian cancer (OC) is the most common cause of death due to gynecologic malignancies, with an estimated five-year survival rate of 39% [1], which varies according to stage of presentation (75% in stage I, 17–20% in stage IV) [2].

OC tends to spread and grow early within the peritoneal cavity and generally occurs at an advanced stage (stages III–IV at time of diagnosis: 70% of patients with OC at the time of diagnosis already have peritoneal disease) [3] (Table 1).

PC is defined as the implantation of neoplastic cells in the peritoneal cavity and is a relatively frequent condition in the advanced stages of many neoplasms of the digestive system (colon, stomach, pancreas) [4]. 

The onset of carcinomatosis represents the most striking picture of the loco-regional spread of the disease, configuring stage III, associated with progression of disease and poor prognosis, causing more than 80% of deaths.

Because of this spread, often the most common symptom is neoplastic ascites, the result of fluid production by peritoneal cells and production of serum and mucin by cancer cells [5].

The treatment plan for patients with OC varies depending on the stage of presentation, and the clinical course is usually characterized by surgery and multiple chemotherapy strategies combined [6].

In patients diagnosed with early or advanced-stage OC, the established treatment protocol involves optimal cytoreductive surgery (CRS) aiming to achieve minimal residual tumor size (less than 1 cm) and subsequent platinum-based chemotherapy. This combination represents the standard of care and is the foremost factor influencing clinical outcomes.

CRS embodies the principle of surgical thoroughness, striving for the complete elimination of macroscopically visible disease or any minute millimeter-sized residue [7]. 

After initial management for late-stage disease, most patients will achieve remission. However, the risk of recurrence remains high (>70–80%), often occurs within 18 months of treatment and is linked to post-surgical residues of the disease and to the chemosensitivity of the tumor; the main site of recurrent disease is the peritoneal cavity [8].

An accurate staging is mandatory to assess the spread of disease and to select patients with resectable peritoneal disease for whom a complete cytoreduction can be achieved.

The peritoneal cancer index (PCI) is a preoperative score system that considers the size and distribution of implants on the peritoneal surface and is the most widespread and comparable carcinomatosis staging system [9].

Despite imaging advancement, diagnostic laparoscopy is still the gold standard for quantifying peritoneal disease [10] by allowing an estimate of tumor extent by PCI.

Multidetector computed tomography (MDCT) is the pillar for the initial staging of patients with OC: it allows the evaluation of the primary tumor, the identification of peritoneal implants and the evaluation of distant metastases both in lymph nodes and in solid organs [11,12,13].

MRI, due to its high contrast resolution, has an accuracy similar to that of CT in staging and even higher than CT in the identification of small peritoneal implants (even without ascites) or in the evaluation of the local extent of the disease. Its usefulness remains, however, limited to selected cases, due to lower spatial resolution, long duration, lower availability, and higher costs of the investigation. 18F-fluorodeoxyglucose positron emission tomography-computed tomography (18FDG PET-CT) is not usually recommended for staging purposes. Its use may be helpful during follow-up in case of suspected recurrence, if CT and MRI results are negative and tumor markers are increased [14].

The optimal treatment in patients with OC is represented by surgical cytoreduction and adjuvant platinum-based chemotherapy, with potential subsequent maintenance strategies established on the basis of two different biomarkers (BRCA status and homologous recombination deficiency status) [15].

The purpose of this review is, therefore, to provide an overview of peritoneal carcinosis in patients with OC, and critically evaluate the most frequent patterns, the routes of diffusion and the sites that are difficult to identify for the radiologist, with particular attention to the role of imaging (CT, MRI, PET-CT) for correct staging. Additionally, the increasing role of radiomic analysis in the staging of patients with advanced OC will be investigated, as a predictive factor in surgical resectability and in the prognosis.

## 2. Materials and Methods

As of September 2023, a structured search was performed using the PubMed database that included all relevant original articles about peritoneal carcinomatosis in OC. No start date limits or language restrictions were used; the research was expanded by also checking the references of the recovered articles for further potentially eligible studies. The search terms consisted of (ovarian cancer) AND (peritoneal carcinomatosis) AND (survival) OR (peritoneal carcinomatosis index) OR (radiomics) OR (treatment). Data mining was performed independently by two reviewers, and any disagreement was discussed with a third auditor.

From the set of articles selected during our literature search, we specifically chose to include studies that offered insights into pathology and imaging techniques. Any editorial comments, conference abstracts and short communications were omitted.

After conducting an initial assessment based on factors such as title, topic, and methodology, articles that were not in line with the objective of our review were excluded. Subsequently, those containing subjective opinions, personal perspectives or anecdotal content were also eliminated.

Our initial literature search yielded 150 articles. Following the application of our predefined criteria, 65 articles were excluded from the review. Ultimately, 87 published articles met the requirements for inclusion in this review [1,2,3,4,5,6,7,8,9,10,11,12,13,14,15,16,17,18,19,20,21,22,23,24,25,26,27,28,29,30,31,32,33,34,35,36,37,38,39,40,41,42,43,44,45,46,47,48,49,50,51,52,53,54,55,56,57,58,59,60,61,62,63,64,65,66,67,68,69,70,71,72,73,74,75,76,77,78,79,80,81,82,83,84,85,86,87] (Figure 1).

## 3. Surgical Staging and Treatment

The primary objective of cytoreductive surgery is to resect all macroscopic tumors or at least to reduce the largest tumor residuals to less than a centimeter. 

When obtaining a complete cytoreduction is technically not feasible due to the spread of disease, or when a patient is unable to tolerate an extensive surgery, then interval debulking surgery (IDS) could be an alternative. IDS implies three cycles of neoadjuvant chemotherapy (NACT) followed by cytoreductive surgery and a further three cycles of adjuvant chemotherapy eventually followed by maintenance treatment based on the molecular characteristics of the tumor. Residual disease at the end of surgery is still considered as the most important prognostic factor impacting survival of patients affected by advanced OC [16,17].

The benefit to survival obtained by a neoadjuvant chemotherapeutic approach followed by cytoreductive surgery compared to primary debulking surgery is still debated. 

The EORTC55971 trial [18] and the CHORUS trial [19] reported analogous values in both similar progression-free survival (PFS; 12 months) and over-all survival (OS; 29 vs. 24 months) for patients affected by advanced OC receiving neoadjuvant chemotherapy and interval debulking surgery compared with upfront debulking surgery. However, in both of these studies, the percentage of patients with complete upfront debulking surgery was very low (<20%). For this reason, a Trial on Radical Upfront Surgical Therapy (TRUST), requiring a qualification process for participating centers in order to reduce an eventual variability of the surgical outcomes, is currently ongoing and expected to conclude in 2024. 

In this scenario, identifying those criteria leading to an optimal cytoreduction is a very important issue. 

According to the most recent ESGO guidelines [14], patients affected by advanced OC are not candidates for primary surgery if the following spread of disease is present:Diffuse deep infiltration of the root of small bowel mesenteryDiffuse carcinomatosis of the small bowel involving such large parts that resection would lead to a short bowel syndrome (remaining bowel < 1.5 m)Diffuse involvement/deep infiltration of:
➢stomach/duodenum;➢head or middle part of pancreas
Involvement of coeliac trunk, hepatic arteries, left gastric artery, Hepatic hilum infiltration or hepatic metastasesMultiple parenchymal lung metastases (preferably histologically proven)Non-resectable lymph nodesBrain metastases

Evidence-based standardized evaluation of the disease extent and of patient condition are essential to predict the possibility of residual macroscopic disease after upfront debulking surgery. Particularly, specific clinical factors (e.g., comorbidities, age, World Health Organization performance status WHO—PS) should also be considered in the pre-operative assessment of operability.

Regarding the spread of the disease, imaging strategies, serum markers and staging, surgical approaches have all been investigated for the prediction of cytoreduction both at primary debulking surgery and at interval debulking surgery, with variable results. 

Diagnostic laparoscopy can provide a definitive histopathological diagnosis and detailed information about the intra-abdominal disease burden [20,21]. 

In a prospective study involving 113 patients with advanced OC, Fagotti et al. [21] investigated by laparoscopy the presence of omental cake, peritoneal and diaphragmatic extensive carcinosis, bowel and stomach infiltration, mesenteric retraction, and spleen and/or liver superficial metastasis. For each patient, a laparoscopic evaluation was conducted, and the total predictive index value (PIV) was computed by aggregating the scores associated with all parameters.

The overall accuracy rate of the laparoscopic procedure to predict an optimal debulking ranged between 77.3 and 100%. The authors observed that when the PIV was greater than or equal to 8, the probability of optimally resecting the disease by laparotomy was equal to 0, and the rate of futile exploratory laparotomy was 40.5%. Vizzielli et al. [22] stratified patients into three groups based on the volume of the disease: high tumor load (HTL) for PIV ≥ 8, intermediate tumor load (ITL) for PIV equal to 6 and 4, and low tumor load (LTL) for PIV < 4, showing that tumor spread according to PIV was an independent prognostic factor impacting either PFS and OS, together with residual tumor (RT) and performance status.

Some other trials investigated the role of diagnostic laparoscopy, with the Fagotti score being still the most adopted in clinical practice [23,24].

In summary, diagnostic laparoscopy is an accurate tool for staging in patients with advanced OC. Some limitations have been noted, including the inability to palpate the liver surface and diaphragm, challenges in visualizing certain anatomical sites due to adhesions in the upper abdomen [25] or in detecting implants of carcinomatosis beyond the gastrosplenic ligament, in the lesser sac, mesenteric root and in the retroperitoneum [23].

Moreover, the presence of huge, fixed masses in the Douglas pouch can reduce the performance of diagnostic laparoscopy. Despite its overall safety, minor complications have been reported. In a study including a series of 145 patients, only one patient showed a serosal injury that was immediately treated [26].

## 4. Imaging

Pre-operative imaging strategies, including CT, MRI, and 18FDG PET-CT, have been largely tested to assess the extent of disease. 

However, the best option technique for staging OC and for detecting all of the implants of PC to assess the non-resectability of the disease does not exist yet.

Among all of the imaging strategies, CT is the most used for preoperative workup and to predict the likelihood of suboptimal cytoreduction. 

With the aim of predicting surgical outcome, some promising imaging models with their related scoring systems have been proposed, however, not yet reproduced for external validation. Several authors have evaluated the use of preoperative CT in assessing tumor resectability with different prediction models. A number of researchers created novel scoring systems that yielded AUC values between 0.67 and 0.97 [27,28], whereas others conducted validation studies on preexisting scoring systems, like the peritoneal cancer index, which produced AUC values in the range of 0.55 to 0.76 [29,30,31].

If identified on CT scan, disease localizations such as diffuse peritoneal thickening, mesenteric disease, suprarenal lymph nodes, ascites, and diaphragmatic or hepatic site disease could be included in potential scoring systems along with clinical features such as age, performance status, and serum tumor markers such as CA 125 value, as they are frequently associated with residual disease.

In recent years, dual energy CT (DECT), a recently introduced technological advancement, has been playing an innovative role by enabling the simultaneous capture of a series of images at different radiant energies during a single CT acquisition [32].

This technique leverages the variations in low-energy and high-energy attenuation of different tissues. In the field of oncology, it enhances the precision and accuracy of identifying neoplasia, allowing for a more targeted approach. Moreover, it offers the possibility of reducing the radiation dose administered to the patient without compromising image quality [32].

Through the utilization of post-processing techniques on DECT data, it becomes feasible to obtain monochromatic images at a specific energy level ranging from 40 to 140 keV.

In the context of advanced OC, this technology could serve as a diagnostic tool for identifying subdiaphragmatic, hepatic, and perisplenic carcinoma implants that may not have been explored surgically [33].

Peritoneal carcinomatosis implants exhibit variable attenuation values, with densitometric characteristics such as those of soft tissue, calcium, and fluid. These characteristics are best visualized through low-kilovolt monochrome images, which highlight contrast differences between tissues at various energy levels. Particularly, DECT excels in detecting implants smaller than 1 cm, proving to be a valuable asset in enhancing the effectiveness of staging for advanced OC, thereby contributing positively to surgical outcomes [33].

However, some of the models reporting good performance for predicting residual disease failed in the external validation phase.

In this context, the external validation of a radiological scoring system appliable for evaluating the spread of the disease, and its diagnostic performance before integration into diagnostic algorithms, is essential.

Finally, recent evidence showed that post-operative imaging-based evaluations of the residual disease differ depending on the intraoperative judgment of the surgeons [34,35].

Particularly, Lorusso et al. [34], analyzed 64 patients with FIGO stage III–IV OC who underwent optimal primary cytoreduction in the same institution with a CT scan performed within 30 days of the surgery. The authors observed that surgeons reported a residual tumor (RT) = 0, 0.1 < RT < 0.5 cm, and 0.6 < RT < 1 cm in 53 (82.8%), 9 (14.1%) and 2 (3.1%) cases, respectively, with postoperative CT scan disagreeing in 13 out of 64 (20.3%) cases. Progression-free survival (PFS) of patients with a positive and negative postoperative CT scan for RT was 5 months (95% confidence interval (CI) 1–15 months) and 28 months (95% CI 2–46 months), respectively (*p* < 0.0001). Evidence of the disease using postoperative CT was an independent prognostic factor in multivariate analysis (hazard ratio (HR) = 8.87, 95% CI = 3.23–24.31, *p* < 0.0001). 

Furthermore, in the study by Heitz et al. [35], the authors hypothesized that an early tumor regrowth might be a contributor to the discordance between surgical assessment and radiologic assessment/integrated assessment. The risk of losing the prognostic factor of complete resection, if chemotherapy is started later than 31 days after primary surgery, was greater than 15%. These data stressed the importance of the timing of chemotherapy initiation (Table 2). 

### 4.1. Computed Tomography

According to the international guidelines, CT represents the imaging modality of choice for OC staging, showing a high accuracy (up to 94%) [11,36].

CT allows evaluation of the extent of the primary tumor, the identification of any peritoneal implants of carcinomatosis and lymph node involvement, and the investigation of the presence of distant metastases (Figure 2).

Strengths of the procedure include wide availability, low cost, high spatial resolution, short scanning time and the possibility of multiplanar image reconstructions (MPRs) [37].

The correct protocol for acquiring CT images to highlight carcinomatosis implants involves the use of intravenous iodine contrast medium (CM) and image acquisition in the portal venous phase (70–90 s) and MPR with a layer thickness of 1–3 mm in multiple planes (axial, coronal and sagittal). Sagittal and coronal reconstructions allow a better evaluation of the subphrenic space and abdominal recesses. Oral CM can be administered to differentiate digestive structures from serous and mesenteric implants [11,38], although it is not currently recommended because it may obscure the presence of calcified peritoneal deposits.

The diagnostic accuracy of CT examination to identify peritoneal carcinomatosis implants is reported to be between 70–90% at all stages of the disease [39].

Considering the heterogeneity of size, morphology, and location of carcinomatosis implants, sensitivity was reported with a wide gap, ranging from 25 to 90% [40,41]. 

Indeed, CT has several disadvantages, such as low soft-tissue resolution, which limits the ability to characterize primary tumors. CT also has limitations in detecting small volume carcinomatosis (<1 cm), especially on the surface of the small bowel or on mesentery root.

In addition, other limitative factors in the detection of carcinosis implants can be the absence of ascites, localization in “challenging” sites such as small bowel, shortage of intra-abdominal adipose tissue and inadequacy of intestinal opacification [38].

Coakley et al. achieved an overall sensitivity of 85–93%, with 25–50% sensitivity in metastases < 1 cm [12], whereas De Bree et al. reported a sensitivity of 9–24% in similarly sized implants [42,43].

Choi HJ et al. showed a lower sensitivity (35.1%) and specificity (68%) for the detection of peritoneal carcinomatosis < 1 cm compared to implants > 1 cm (52.4 and 75%, respectively; *p* = 0.037) [44].

### 4.2. Magnetic Resonance Imaging

MRI is performed in cases where CT examination is contraindicated (pregnant patients or allergies to iodized CM), if the CT findings are inadequate/doubtful for the presence of metastases or if the implants are in sites where CT proves inadequate, such as subphrenic spaces, lesser omentum, serosal, and mesenteric deposits [45].

The MRI study protocol includes T1- and T2-weighted image sequences on multiple planes (axial, sagittal and coronal) with and without adipose tissue signal suppression, DWI on axial plane at least using two b factor (0, 1000 s/mm^2^), over the entire abdomen and pelvis, and dynamic contrast-enhanced MRI after injection of paramagnetic CM.

T2-weighted images and DWI, including apparent diffusion coefficient (ADC) maps, are pivotal to improve the identification of even small peritoneal carcinomatous implants, especially on the mesentery, bowel serosa and peritoneal reflections, due to the significant contrast between the lesion and surrounding peritoneal tissues [45] (Figure 3).

Moreover, involved peritoneal lining may be shown by dynamic contrast-enhanced MRI as a delayed enhancement.

Nevertheless, the usefulness of MRI remains limited by potential artifacts (e.g., motion or magnetic susceptibility artifacts), long duration of the investigation, lower availability, higher costs, long interpretation times and simultaneous analysis of the abdomen and pelvis.

Concerning the sensitivity and specificity of the MRI in detecting peritoneal carcinomatosis implants, Fujii et al. found values of 90% and 95.5%, respectively, with the use of DWI sequences [46].

Yu et al. found that the sensitivity and specificity of MRI for detecting peritoneal deposits in ovarian neoplasm were 88% and 99%, respectively [47].

Compared to CT, MRI demonstrated a superior sensitivity and accuracy (MRI: 95% and 88% vs. CT: 55% and 63%, respectively) thanks to the use of DWI at high b values and the administration of paramagnetic CM [48,49].

Concerning sizes, MRI was shown to have better sensitivity (85–90%) than CT in the detection of implants < 1 cm [50], but little difference was seen in one of the largest series in which the majority of patients (88%) had implants > 2 cm and presence of ascites (sensitivities of MR and CT, respectively, 95% and 92%) [39,51].

In recent years, the development of new techniques, such as whole-body diffusion-weighted imaging (WB-DWI/MRI) improved the diagnostic accuracy. 

Rizzo et al., in a cohort of 92 patients evaluated by CT and WB-DWI/MRI, showed significantly higher accuracy of WB-DWI/MRI specifically for involvement of mesentery, lumbo-aortic lymph nodes, pelvis, large bowel, and sigmoid-rectum [52].

Findings by Michielsen et al. [53] indicated that WB-DWI/MRI outperformed CT in terms of sensitivity (94% vs. 66%), specificity (98% vs. 77%), and accuracy (96% vs. 71%) for detecting disease sites that suggested non-resectability. 

Conversely, when applying the ESMO-ESGO criteria for non-resectability, Fischerova et al. found no statistically significant variations in the outcomes between WB-DWI/MRI, pelvic and abdominal ultrasound, and contrast-enhanced CT when predicting residual disease upon completion of the surgery [54].

### 4.3. PET-CT

According to the European Society for Medical Oncology (ESMO), PET-CT is not recommended as an imaging technique for initial management of epithelial ovarian carcinoma [55]. 

The main disadvantage of 18F FDG PET-CT is the limited spatial resolution (5–6 mm) in detection of small-volume carcinomatosis, especially on the small bowel/colon serosa or their mesenteries; moreover, the results may be misinterpreted due to the uptake of 18F FDG caused by physiological movements (e.g., of the digestive tract) or non-malignant and inflammatory lesions, giving rise to false-positive results [56,57,58].

Michielsen et al. found a lower sensitivity of PET-CT in PC detection in small bowel mesentery (33%), colon serosa (27%) and colon mesentery (25%) compared to CT (63%, 45% and 50%, respectively). Specificity was, however, overlapping [53].

Lopez-Lopez et al. compared the 18F FDG PET-CT with CT in 59 patients, showing a sensitivity of 35% and 24%, respectively, whereas CT had higher specificity (98% vs. 93%) [59].

18F FDGPET-CT may still be used for staging as a problem-solving tool if unclear CT findings are detected (such as indeterminate lymph node involvement in the retroperitoneum or mediastinum), providing in a single test, anatomical and functional information of carcinosis implants.

### 4.4. PET-MRI

PET-MRI is an emerging fusion technique that, despite the few studies carried out, has shown important results in the OC characterization thanks to high soft tissue contrast of MRI along with functional imaging of FDG uptake. In a pilot study, compared to DW-MRI, PET-MRI turned out to have higher sensitivity for detection of carcinomatosis in 31 patients with OC, especially in “challenging sites” (three out of four in small bowel regions) [60].

Combined PET-MRI has proven to be helpful in the characterization of ovarian tumors with a sensitivity and specificity of 94% and 100%, respectively, compared to PET-CT (74–80%) and MRI (84–60%) [61].

Although this promising hybrid imaging technique could soon be included for a better evaluation of OC peritoneal carcinomatosis, further investigations are needed for clarification of its role. 

## 5. Diffusion Pathways

Peritoneal extension of OC is considered a negative prognostic factor associated with higher risk of recurrence and high mortality compared to cancers diagnosed at an early stage (I or II) [62].

Spread of ovarian carcinoma to the abdominal cavity usually occurs through the peritoneal circulation [63]. In order to understand how the tumor spreads within the peritoneum, it is necessary to know its anatomy and function. The peritoneum is a serous membrane composed of two layers continuous with each other: the outer parietal layer, lining the abdominal cavity and pelvis, and the inner visceral layer, lining the intraperitoneal visceral organs; the latter reflects and folds to line the visceral organs and keep them suspended in the cavity, thus forming mesenteries, oments and ligaments that divide the abdomen into several compartments. The space between the parietal and visceral layers of the peritoneum is the peritoneal cavity and is filled with a slight amount of fluid, which allows frictionless movement of the visceral organs.

Such spaces and supporting structures can serve as gateways for intraperitoneal tumor spread and the establishment of carcinomatosis implants [37,64].

Peritoneal fluid is not stationary; rather, it follows a dynamic circulation related to diaphragmatic respiratory movements: in the upright position, peritoneal fluid accumulates in most of the declivous portions of the abdomen, such as the recto-uterine and paravesical recesses. Fluid flows from the pelvis to the paracolic gutter, and then to the subdiaphragmatic regions during the expiratory phase whereby the diaphragm moves upward and generates negative intraabdominal pressure, drawing back the fluid in a cranial direction.

On the right side, fluid moves from the paracolic gutter to the anterior subhepatic space and into the right hepatic spaces. On the left side, fluid ascending toward the paracolic gutter is arrested by the left phrenicocolic ligament, so its progression into the perisplenic spaces is confined. In addition, the falciform ligament is an anatomical barrier to fluid progression from the right subdiaphragmatic spaces to the left perisplenic spaces.

The kinetics of intraperitoneal fluid explain why implants of peritoneal carcinomatosis are more often located in the paracolic gutters and right subdiaphragmatic spaces, rather than in the left ones, and on higher constriction sites such as Douglas’ pouch and the right lower quadrant [63,65].

## 6. Disease Patterns

Implants of peritoneal carcinomatosis should be characterized by different morphological and dimensional features since they are extremely important for an ideal presurgical and pretreatment evaluation.

Number (solitary or multiple) and density or intensity with and without intravenous injection of contrast are other parameters to evaluate.

Nodules of peritoneal carcinomatosis could be morphologically divided into solid, cystic and mixed implants with either a solid component or a cystic component, although rarely, mixed solid and cystic or purely cystic lesions are found [4,62] (Figure 4).

Moreover, serous cystadenocarcinoma, an OC subtype, can produce calcified peritoneal metastatic deposits [66].

Some cystic implants are low in attenuation and mimic loculated fluid [67].

Micronodular pattern refers to milky spots of peritoneal implants smaller than 5 mm involving the parietal or visceral peritoneum and mesenteric fat; on the contrary, nodular pattern is characterized instead by oval shape implants or coalescing small lesions (>5 mm) diffusely involving the tunica serosa and mesenteric, sometimes presenting spiculated margins.

Micronodular patterns observed in the mesentery can appear as thickening of the root with a stellate pattern [67].

Nodular lesions coalescing in irregular soft-tissue thickenings of variable extension that coat the viscera refer to plaque-like patterns. This type of lesion is typically found in the subdiaphragmatic spaces involving liver and spleen surfaces and presenting lower attenuation than the parenchyma on contrast-enhanced scans [68].

Large plaques involving omental fat and surrounded by reactive fibrotic tissue are referred to as “omental cakes” (Figure 5).

Implants of several centimeters, resulting from the confluence of smaller nodules, can lead to soft-tissue masses (mass-like pattern), usually found in the pelvis; Masses measuring 10 cm or larger are called “bulky tumor” [4].

Subcutaneous nodules in the anterior abdominal wall may sometimes be the first clinical manifestation of OC (Sister Mary Josef’s nodules). They are typically found in the periumbilical zone and can be direct extensions of omental disease [69].

Diffusely infiltrating tumor or focal soft-tissue masses on the bowel surface and mesentery can tether the loops and straighten the mesenteric vasculature, eventually causing bowel obstruction and dilatation of proximal loops (ileal freezing) [4,68].

## 7. Scoring System in Diagnostic Imaging 

### 7.1. Peritoneal Cancer Index (PCI)

The peritoneal cancer index, adapted for imaging, is the only externally validated system. With the aim of creating a peritoneal evaluation system useful in the concise, clinically relevant, and statistically assessable preoperative and follow-up setting, Sugarbaker devised the PCI, a scoring system determined by the distribution and size of the tumor within the abdominopelvic cavity found on direct examination or through CT. 

Tracing two sagittal lines and two transverse lines divides the abdomen into nine abdominopelvic regions that are numbered from 0 to 8, starting from the umbilical region and proceeding clockwise. The small intestine, unlike the large intestine that is evaluated in the respective abdominal regions 0–9, is evaluated separately and is divided into four further regions called 9 to 12 (9: upper jejunum; 10: lower jejunum; 11: superior ileum; 12: inferior ileum). For each of these regions, the volume of the tumor that occupies it is then indicated: V0 indicates the absence of tumor localization in the abdominopelvic region described; V1 indicates the presence of nodules with a diameter < 0.5 cm; V2 indicates nodules with a diameter between 0.5 and 5 cm; V3 indicates nodules with a diameter > 5 cm [70] (Figure 6).

Evaluation of the correlation between preoperative CT-PCI and surgical outcome and overall survival in patients with epithelial OC demonstrated that preoperative CT-PCI correlates with the probability of post-operative residual disease in patients undergoing primary cytoreduction. In addition, it showed that the serous histotype is significantly associated with higher CT-PCI scores and that it has higher prevalence in the upper abdominal and intestinal regions than in the other histotypes [31]. 

PCI is considered a prognostic indicator of survival in OC. Patients with PCI < 10 show better survival than those with PCI > 10, and even excluding stage IV patients from the analysis, PCI remains a significant survival index. Patients with PCI > 10 do not have prolonged survival and are, therefore, considered a high-risk group even if they have performed complete or near complete cytoreduction and standard treatment with systemic chemotherapy [71]. In patients with advanced epithelial OC, it has been proposed to evaluate PCI only in the regions corresponding to the small intestine and the hepatoduodenal ligament (9–12 + 2), as it has been demonstrated that they are more predictive for a complete resection and for survival based on the sum of the total PCI [72]. 

Evaluation of the prognostic value of small bowel PCI in patients with advanced epithelial OC undergoing cytoreduction and hyperthermic intraperitoneal chemotherapy (HIPEC) shows that both small bowel PCI and cytoreduction completeness are independent prognostic factors of overall survival, while age and timing of HIPEC have not been identified as independent prognostic factors [73]. 

Currently a clinical trial, “Imaging Study in Advanced ovArian Cancer (ISAAC)”, is investigating the diagnostic performance of the peritoneal cancer index using ultrasound, WB-DWI/MRI and CT [29].

On the basis of previous results comparing imaging methods with the surgical approach for PCI, CT-PCI showed lower accuracy than surgical PCI in both high- and low-volume patients of disease. The difference in CT-PCI compared to surgical PCI is significant both in patients with OC and in patients treated with neoadjuvant chemotherapy for peritoneal disease [74]. Mikkelsen et al. [75] compared the efficacy of DW-MRI, CT and FDG PET/CT in PCI vs. surgical assessment. The mean surgical PCI was 18 (range 3–32), and all three imaging modalities often underestimated surgical PCI with a mean difference from surgical PCI of 4.2 (95% CI: 2.6–5.8) for CT, 4.4 for DW-MRI (95% CI: 2.9–5.8) and 5.3 for FDG PET/CT (95% CI: 3.6–7.0) in the absence of statistically significant differences between the three different imaging modalities.

A PCI > 20 evaluated by laparotomy and an albumin concentration < 33 g/L can predict the onset of high-grade complications after OC surgery. The main high-grade complication (28/62 patients—45.2%) in these patients was pleural effusion [76] (Table 3). 

### 7.2. Bowel, Upper Abdomen, Mesentery in Peritoneal Metastasis (BUMPY)

Recently, another score has been proposed by Nougaret et al. [7] on the basis of radiological criteria to assess resectability in OC. Nougaret collected such planting sites under the acronym BUMPy (Bowel, Upper abdomen, Mesentery in Peritoneal metastasis). It is well known that some sites of carcinomatosis correlate with a suboptimal cytoreduction or require particular attention on the surgical level. Regarding localizations, peritoneal implants are divided into resectable and unresectable. Resectable implants are classified into implants with limited involvement of the small bowel (few nodules with serous involvement and nodules on the antimesenteric side) and of the mesentery (scattered nodules). Unresectable implants are classified into implants with diffuse involvement of the mesentery (many nodules, retractile and infiltrative pattern) and of the small bowel (tumor-like pattern, and both serosa and adjacent mesentery involved in multiple segments). The author suggests analyzing the images following the direction of peritoneal flow. The analysis is carried out on the coronal plane starting from the pelvis describing the involvement of the Douglas cord and then moving up right towards the paracolic gutter, the serous membrane of the ascending colon and the Morrison pouch. It continues with the evaluation of the hepatic capsule and the right hemidiaphragm. This is followed by the evaluation of the gastrohepatic ligament, the serous of the transverse mesocolon and the gastric ligament, then the left hemidiaphragm, the gastrosplenic ligament, the spleen, the descending colon and the left paracolic gutter. Finally, the mesentery is evaluated. At the end of the analysis on the coronal plane, it is suggested to repeat the evaluation on the axial plane (Figure 7).

## 8. Radiomics

The process of radiomics analysis is based on established steps, each one in continuous evolution over time thanks to technological and mathematical advances, and they are the same for all radiomics studies, independently from the anatomy, pathology, and outcomes under examination [77,78,79]. The main steps are image acquisition and segmentation, feature extraction, feature selection and model construction [80].

Although some studies have so far evaluated radiomics and radiogenomics of OC, only a few of them have evaluated the possibility to predict the cytoreduction.

Thanks to advancements in The Cancer Genome Atlas (TCGA), a prognostic algorithm for high-grade serous OC has been defined with four different subtypes: differentiated, immunoreactive, mesenchymal, and proliferative. Vargas et al. explored the relationships between subjective qualitative CT features and the different subtypes of OC, showing that the mesenchymal subtype was significantly associated with higher risk of peritoneal involvement and the presence of mesenteric infiltration on CT [81], which is considered one of the reasons for failure of cytoreduction. In a different study including 38 patients, 12 quantitative metrics were selected to represent the inter-site imaging heterogeneity, and these metrics were associated with incomplete surgical resection (similarity -level cluster shade, inter-site similarity-level cluster prominence, and inter-site cluster variance) [82].

Rizzo et al. evaluated CT radiomics features in 101 patients, extracted from the primary tumor alone and combined with clinical data, showing that radiomic features related to mass size, randomness and homogeneity were associated with residual tumor at surgery [83], which still represents the most important feature for a complete cytoreduction and for prognosis. 

Meier et al. assessed associations between inter-site texture heterogeneity parameters derived from CT, survival, and BRCA mutation status in 88 OC patients. They showed that high values of the three metrics used for the model were significantly associated with lower complete surgical resection status in BRCA-negative patients, but not in BRCA-positive patients, although the model was not able to distinguish the presence or absence of BRCA mutation [84].

More recently, studies based on MRI are underway in evaluating radiomic features in OC. To this end, Yu et al. assessed MR radiomic features in 86 patients with OC with the aim of predicting the peritoneal carcinomatosis. The authors showed that the radiomics nomogram constructed by combining radiomics characteristics and clinicopathological risk factors showed a better diagnostic effect than the clinical model and the radiomics model alone [85]. Likewise, in a recent study, Song et al. generated a radiomic signature based on MRI features to predict the presence of peritoneal carcinomatosis before surgery in 89 patients. The nomogram, comprising the radiomics signature (based on six features), pelvic fluid, and CA-125 level, showed the best discrimination with an AUC of 0.969 in the training cohort and 0.944 in the validation cohort [86].

Although there is strong interest in radiomics for the prediction of peritoneal carcinomatosis and prognosis in OC patients, there are currently many tools based on artificial intelligence that do not include imaging data, thus showing the gap that still exists in this field. In the future, more precise descriptions of the methods and integration of multi-omics models may lead to an out-performance of single-omic datasets [87], offering adjunctive help for prognostication and treatment planning for OC patients (Table 4).

## 9. Conclusions

The high aggressiveness of OC leads to frequent peritoneal dissemination. Utilizing various imaging techniques, including CT, MRI, and PET-CT, plays a crucial role in guiding the diagnostic workflow for patients due to their sensitivity and specificity in identifying various morphological patterns and disease spread pathways. Disease staging systems like PCI and its derivatives establish a threshold for directing treatment decisions and assessing prognosis in patients. Radiomics may serve a significant role in identifying suitable candidates for surgical treatment and predicting optimal cytoreduction. Moreover, it has the potential to aid in prospective risk stratification for PC, showing promise as a valuable addition, though further research in this area is currently required.

## Figures and Tables

**Figure 1 cancers-15-05827-f001:**
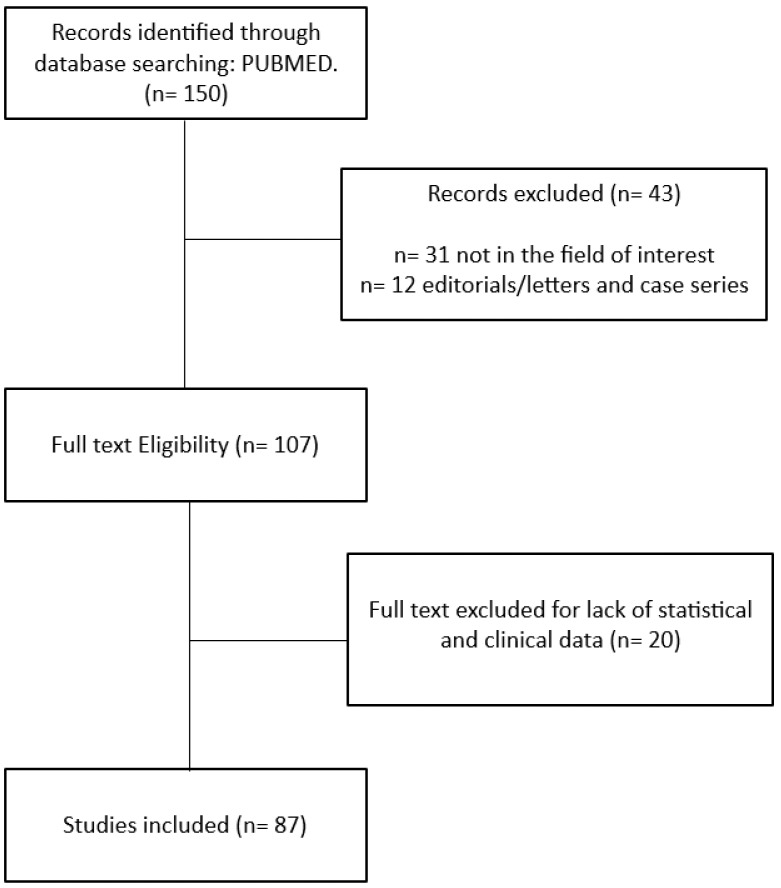
Study selection flow chart.

**Figure 2 cancers-15-05827-f002:**
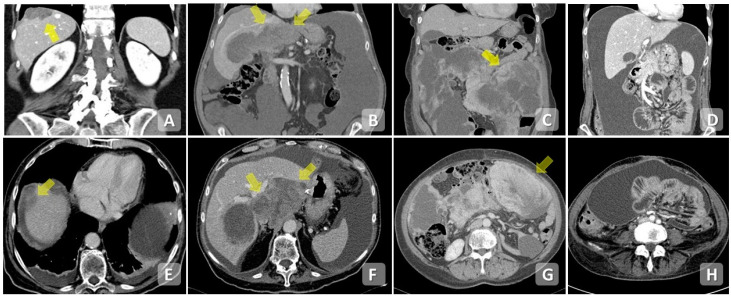
Contrast-enhanced CT scan, axial planes (**E**–**H**) and MPRs (**A**–**D**). Extensive infiltration of the diaphragmatic dome (**A**,**E**) and hepatic hilum (**B**,**F**). Diffuse mesenteric infiltration (**C**,**G**,**D**,**H**). Note the presence of free peritoneal fluid and pleural effusion.

**Figure 3 cancers-15-05827-f003:**
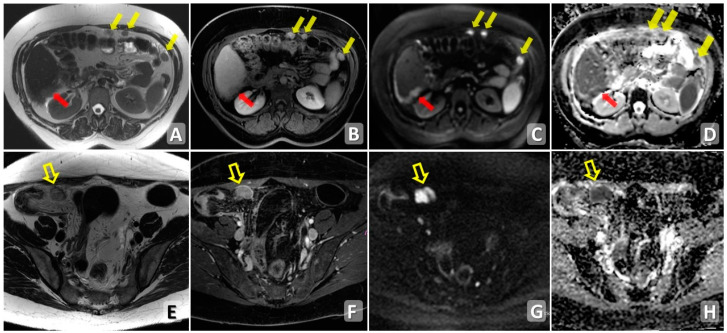
MRI axial T2WI (**A**,**E**), post-contrast fat-suppressed T1WI (**B**,**F**), DWI (**C**,**G**) and ADC map (**D**,**H**) showing multiple centimetric nodules of peritoneal carcinosis. Upper row: multiple nodules of PC (yellow arrows) showing post-contrast enhancement (**B**) and signal restriction in DWI/ADC (**C**,**D**). Additionally, a plaque of PC is localized on the Glissonian surface well recognizable in DWI (red arrow) at high b value (**C**,**D**). Lower row: macronodule of PC in the right iliac fossa (empty arrows) showing post-contrast enhancement (**F**) and signal restriction in DWI/ADC (**G**,**H**).

**Figure 4 cancers-15-05827-f004:**
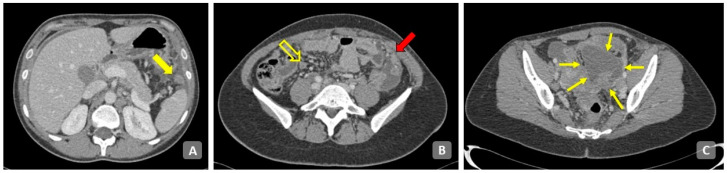
Axial contrast-enhanced CT of the abdomen. (**A**) Upper abdomen. Perisplenic carcinosis with micro and macronodular pattern (solid arrow). (**B**) Close to the ileocecal junction (empty arrow) an inhomogeneous density of adipose tissue is present with nodules and septa as reticular-nodular pattern of PC. Omental cake is present in the left side (arrow). (**C**) Ovarian mass with mixed solid/cystic components (small arrows). Diffuse infiltration of the sigmoid colon is present.

**Figure 5 cancers-15-05827-f005:**
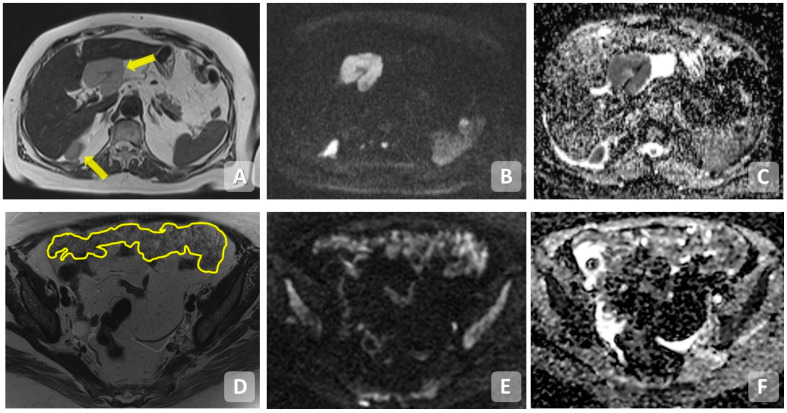
MRI images of peritoneal carcinomatosis. Upper row: (**A**) Axial T2WI, (**B**) DWI, (**C**) ADC—carcinomatosis nodule in perihilar and posterior pericapsular hepatic area (arrows); lower row, (**D**) T2WI, (**E**) DWI, (**F**) ADC—carcinomatosis with omental cake pattern (underlined).

**Figure 6 cancers-15-05827-f006:**
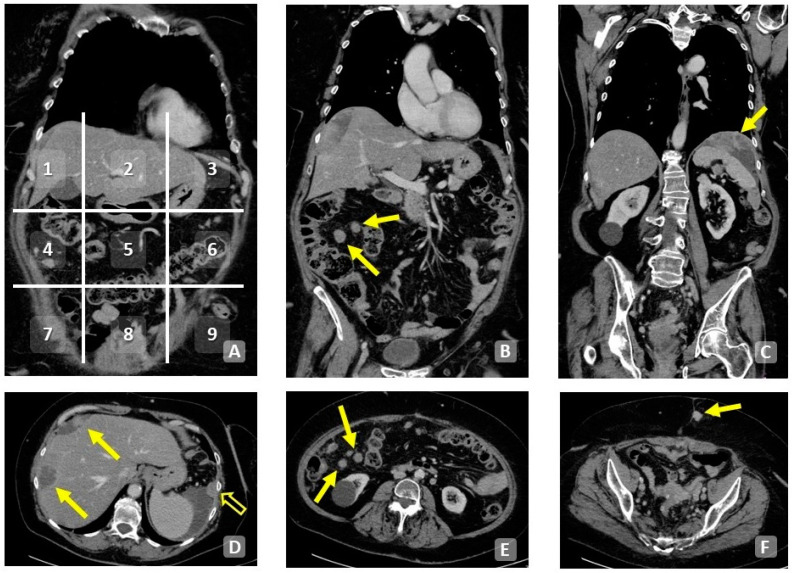
A 73-year-old patient. Coronal (**A**–**C**) and axial (**D**–**F**) contrast-enhanced CT scan of the abdomen. Division into quadrants of the abdomen with PCI numbering (**A**). Macronodules of peritoneal carcinomatosis in the context of the mesentery (arrows, (**B**,**E**)). Left subphrenic plaque of peritoneal carcinomatosis (**C**). Multifocal extensive infiltration of the hepatic surface (**D**). Subcutaneous implantation of peritoneal carcinomatosis (**E**).

**Figure 7 cancers-15-05827-f007:**
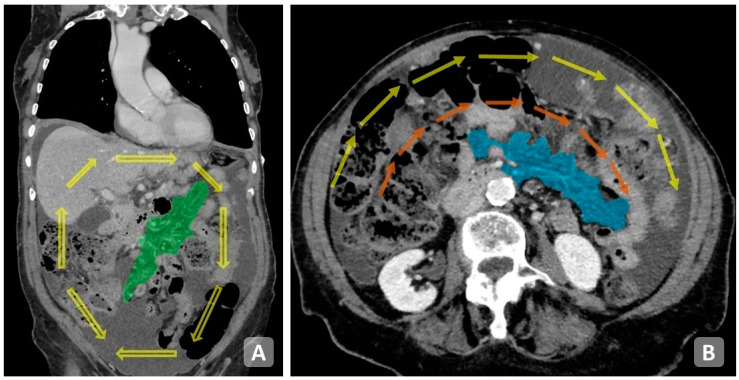
Coronal plane showing peritoneal fluid flow direction (arrows) and mesentery (green area) (**A**). Axial plane showing suggested evaluation direction (double wagon wheel-like) and mesentery (blue area) (**B**).

**Table 1 cancers-15-05827-t001:** Figo staging of ovarian cancer.

FIGO staging of ovary cancer	**Stage**	**Description**
	I	Tumor confined to the ovaries or fallopian tube(s) IA: Limited to one ovary (capsule intact) or fallopian tube IB: Limited to both ovaries (capsule intact) or fallopian tubes IC: Limited to one or both ovaries or fallopian tubes, with any of the following: IC1: Surgical spill intraoperatively IC2: Capsule ruptured before surgery, or tumor on ovarian or fallopian tube surface IC3: Malignant cells present in the ascites or peritoneal washing
	II	Tumor involves one or both ovaries or fallopian tubes or is primary peritoneal cancer and involves other pelvic organs IIA: Extension and/or implants on the uterus and/or fallopian tubes and/or ovaries IIB: Extension to the other pelvic intraperitoneal tissue
	III	Tumor involves one or both ovaries or fallopian tubes or primary peritoneal cancer and spreads beyond the pelvis but not outside the abdominal cavity IIIA: Cancer involves the pelvic structures and the retroperitoneal lymph nodes, without macroscopic visible tumor outside of the pelvis IIIB: Cancer involves structures outside of the pelvis (<2 cm) IIIC: Cancer involves structures outside of the pelvis (>2 cm). This included surface implants along abdominal solid organs, without parenchymal involvement.
	IV	Distant metastasis excluding peritoneal metastases IVA: Metastatic pleural effusion IVB: Parenchymal metastatic lesion and/or metastases to extra-abdominal organs (including inguinal and thoracic lymph nodes)

**Table 2 cancers-15-05827-t002:** Imaging study results.

	Modality	Authors	Title	Patients	Aim of Study	Sensitivity (%)	Specificity (%)
1	CT	Choi H.J. et al., 2010 [44]	Region-based diagnostic performance of multidetector CT for detecting peritoneal seeding in OC patients	57	To determine the accuracy of CT compared with the surgical findings (peritoneal seeding, metastatic lymph nodes) in OC patients	45	72
2	CT MRI	Tempany C.M.C. et al., 2000 [39]	Staging of advanced OC: comparison of imaging modalities-report from the radiological diagnostic oncology group	118	To compare multiple imaging modalities for diagnosing and staging advanced OC	92 95	82 80
3	CT	Mazzei M.A. et al., 2013 [43]	Accuracy of MDCT in the preoperative definition of peritoneal cancer index (PCI) in patients with advanced OC who underwent peritonectomy and hyperthermic intraperitoneal chemotherapy	43	To assess MDCT accuracy in preoperatively defining the peritoneal cancer index (PCI) in individuals with advanced ovarian cancer	100	40
4	CT MRI	Qayyuma A. et al., 2004 [41]	Role of CT and MR imaging in predicting optimal cytoreduction of newly diagnosed primary epithelial OC	137	To ascertain the comparative precision of CT and MR imaging in identifying non-surgically manageable tumor sites before cytoreductive surgery in patients with primary ovarian cancer	79 71	99 100
5	MRI	Ricke J. et al., 2002 [51]	Prospective evaluation of contrast-enhanced MRI in the depiction of peritoneal spread in primary or recurrent OC	57	To evaluate MRI accuracy in the staging of intra-abdominal tumor dissemination in ovarian cancer	90.9	57.1
6	PET/CT CT	Kim H.W. et al., 2013 [58]	Peritoneal carcinomatosis in Patients with OC—Enhanced CT Versus 18F-FDG PET/CT	46	To conduct a comparative analysis of the diagnostic accuracy between FDG PET/CT and enhanced abdominal CT	96.2 88.5	90 65
7	WB-DWI/MRI	Michielsen K. et al., 2014 [53]	Whole-body MRI with diffusion-weighted sequence for staging of patients with suspected ovarian cancer: a clinical feasibility study in comparison to CT and FDG-PET/CT	32	To evaluate whole-body DWI/MRI diagnostic effectiveness in staging and determining operability, in contrast to CT and FDG-PET/CT, for individuals with suspected ovarian cancer	91	91
8	CT F-FDG PET/CT	Lopez-Lopez V. et al., 2016 [59]	Use of (18)F-FDG PET/CT in the preoperative evaluation of patients diagnosed with peritoneal carcinomatosis of ovarian origin, candidates to cytoreduction and hipec. A pending issue	59	To evaluate the clinical usefulness of the results obtained with 18F-FDG PET/CT in relation to CT in the preoperative staging of patients with peritoneal carcinomatosis secondary to primary or recurrent OC	35 24	98 93

**Table 3 cancers-15-05827-t003:** PCI study results.

	Author	Title	Patients	Results
1	Rosendahl M, et al. (2018) [72]	Specific regions, rather than the entire Peritoneal Carcinosis Index, are predictive of complete resection and survival in advanced epithelial ovarian cancer	673	The predictive value of complete resection and survival is higher when specific PCI regions related to the small intestine and hepatoduodenal ligament are chosen compared to considering the entire PCI.
2	Tentes A.-A. K, et al. (2003) [71]	Peritoneal Cancer Index: a prognostic indicator of survival in advanced ovarian cancer	60	The extent of peritoneal spread in advanced ovarian cancer can be thoroughly evaluated through the peritoneal cancer index. This index plays a crucial role as a prognostic factor for survival and proves valuable in identifying distinct subgroups.
3	Avesani G, et al. (2020) [31]	Radiological assessment of Peritoneal Cancer Index on preoperative CT in ovarian cancer is related to surgical outcome and survival	297	The evaluation of preoperative CT-assessed PCI is linked to the likelihood of residual disease following cytoreductive surgery. Nevertheless, its effectiveness as a primary screening test to consistently pinpoint patients suitable for complete cytoreductive surgery is limited. CT-PCI exhibits a positive correlation with both disease-free survival and overall survival, thus serving as a potentially valuable independent prognostic factor.
4	Lomnytska M, et al. (2021) [76]	Peritoneal Cancer Index predicts severe complications after ovarian cancer surgery	256	Peritoneal cancer index ≥ 21 was an independent predictor of high-grade complications after ovarian cancer surgery. Increased peritoneal cancer index also impacted overall survival negatively, but high-grade complications did not influence overall survival.
5	Iavazzo C, et al. (2021) [73]	Small Bowel PCI Score as a prognostic factor of ovarian cancer patients undergoing cytoreductive surgery (CRS) with hyperthermic intraperitoneal chemotherapy (HIPEC), a retrospective analysis of 130 patients	130	A statistically significant correlation between small bowel-PCI score and overall survival of patients with advanced ovarian cancer was revealed.
6	Mikkelsen MS, et al. (2021) [75]	Assessment of peritoneal metastases with DW-MRI, CT, and FDG PET/CT before cytoreductive surgery for advanced stage epithelial ovarian cancer	50	None of the imaging modalities, including DW-MRI, CT, and FDG PET/CT, demonstrated superiority in the preoperative evaluation of surgical PCI in patients scheduled for upfront CRS for advanced stage EOC.
7	Goswami G, et al. (2019) [74]	Accuracy of CT scan in predicting the surgical PCI in patients undergoing cytoreductive surgery with/without HIPEC-a prospective single institution study	50	CT-PCI shows lower accuracy than surgical PCI in both high- and low-volume patients of disease. The difference in CT-PCI compared to surgical PCI is significant both in patients with ovarian cancer and in patients treated with neoadjuvant chemotherapy for peritoneal disease.

**Table 4 cancers-15-05827-t004:** Radiomics study results.

	Author	Title	Patients	Results
1	Vargas HA, et al. (2018) [81]	Radiogenomics of High-Grade Serous Ovarian Cancer: Multireader Multi-Institutional Study from the Cancer Genome Atlas Ovarian Cancer Imaging Research Group	92	Combinations of imaging features contained predictive signal for time to progression and CLOVAR profile. Interobserver agreement was strong for some features, but could be improved for others.
2	Vargas HA, et al. (2017) [82]	A novel representation of inter-site tumour heterogeneity from pre-treatment computed tomography textures classifies ovarian cancers by clinical outcome	38	Of the 12 inter-site texture heterogeneity metrics evaluated, those capturing the differences in texture similarities across sites were associated with shorter overall survival and incomplete surgical resection.
3	Rizzo S et al. (2018) [77]	Radiomics of high-grade serous ovarian cancer: association between quantitative CT features, residual tumour and disease progression within 12 months.	101	This study found significant associations between radiomic features and prognostic factors, such as residual tumour and progressive disease at 12 months
4	Meier A et al. (2019) [84]	Association between CT-texture-derived tumor heterogeneity, outcomes, and BRCA mutation status in patients with high-grade serous ovarian cancer.	88	Higher inter-site cluster variance was associated with lower PFS (*p* = 0.006) and OS (*p* = 0.003). Higher inter-site cluster prominence was associated with lower PFS (*p* = 0.02) and higher inter-site cluster entropy (SE) correlated with lower OS (*p* = 0.01). High values of the three metrics were significantly associated with lower complete surgical resection status in BRCA-negative patients
5	Yu XY et al. (2021) [85]	Multiparameter MRI Radiomics Model Predicts Preoperative Peritoneal Carcinomatosis in Ovarian Cancer	88	The radiomics model from the multiparametric-MRI combined sequence showed a higher area under the curve than the model from FS-T2WI, DWI, and DCE-MRI alone. A radiomics nomogram constructed by combining radiomics features and clinicopathological risk factors showed a better diagnostic effect than the clinical model and the radiomics model.
6	Song XL et al. (2021) [86]	Radiomics based on multisequence magnetic resonance imaging for the preoperative prediction of peritoneal metastasis in ovarian cancer.	89	The radiomics signature generated by 6 selected features showed a favorable discriminatory ability to predict peritoneal metastases. The nomogram, comprising the radiomics signature, pelvic fluid, and CA-125 level, showed more favorable discrimination.

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
