# Peer review of "Imaging of Peritoneal Carcinomatosis in Advanced Ovarian Cancer: CT, MRI, Radiomic Features and Resectability Criteria"

_cancers, 2023, doi:10.3390/cancers15245827_

Round 1

Reviewer 1 Report

Comments and Suggestions for Authors

The review developed by Miceli et al. is based on 76 references published between 1995 and 2023 years. 

The authors structured the paper on different acquisition ways e. g. CT, PET-CT, or MRI.

The structure and conceptualization have some drawbacks.

1. From the title “Imaging of peritoneal carcinosis in advanced ovarian cancer:  CT, MRI, radiomic features and resectability criteria” it is found that the acquisition methods are CT and MRI, instead the paper deals with additional MRI . 

2. It is unclear the criteria for grouping references in sections: please add these in the Introduction in the last paragraph these specifications.

3. Please add a section whose database belongs to the images that are shown in Figures 1-6.

4. The acquisition stage supposes a lot of noise having different types, it is unclear if the studies taken into account in this paper this aspect. 

5. In the section „Radiomics.”, please add a table as in the previous sections. 

The titles of the table are e.g.  „PCI study conclusions” but these do not contain only conclusions 

Minor observation 

In the paper unjustified spaces are between lines e. g . 78-79, 202-204, 

After the title please do not add  the point e. g.  „Radiomics.”

#lines 509-523 do not respect the template

It is difficult to understand the structure of the paper because the title and subtitle are not numbered.

Author Response

  1. 1. From the title “Imaging of peritoneal carcinosis in advanced ovarian cancer: CT, MRI, radiomic features and resectability criteria” it is found that the acquisition methods are CT and MRI, instead the paper deals with additional MRI .

Thanks for your suggestion. In this review we have chosen to deal mainly with CT and MRI imaging methods

  1. 2. It is unclear the criteria for grouping references in sections: please add these in the Introduction in the last paragraph these specifications.

Table 2, table 3 and table 4 group the results of the studies used in the bibliography to write the different sections of the main text.

  1. 3. Please add a section whose database belongs to the images that are shown in Figures 1-6.

Images in the main text are all from our patients in Sapienza University.

  1. 4. The acquisition stage supposes a lot of noise having different types, it is unclear if the studies taken into account in this paper this aspect.

Thanks for the suggestion. If you're referring to the article selection phase, having chosen to conduct a narrative review we chose to explore the topic of ovarian carcinosis as much as possible and then make a selection of the collected material.

  1. 5. In the section „Radiomics.”, please add a table as in the previous sections.

Thanks for the suggestion. Done.

The titles of the table are e.g.  „PCI study conclusions” but these do not contain only conclusions

Thanks for the suggestion. We changed tables’ title in “studies results”

Minor observation

In the paper unjustified spaces are between lines e. g . 78-79, 202-204,

After the title please do not add  the point e. g.  „Radiomics.”

#lines 509-523 do not respect the template

It is difficult to understand the structure of the paper because the title and subtitle are not numbered.

Thanks for the suggestion. We have corrected the mistakes.

Reviewer 2 Report

Comments and Suggestions for Authors

I have a few minor comments on this well done, comprehensive and exhaustive review manuscript on imaging of peritoneal carcinosis in patients with advanced ovarian cancer:

-The Introduction should be shortened by at least 30% and the various statements should be linked together more organically. This would allow focusing more on the purpose of the review and going straight to the study findings.

-Regarding CT imaging, I suggest mentioning the added value of dual energy CT over conventional CT for the detection and characterization of peritoneal deposits in patients with advanced OC.

Comments on the Quality of English Language

Some minor editing of the English language would be required.

Author Response

I have a few minor comments on this well done, comprehensive and exhaustive review manuscript on imaging of peritoneal carcinosis in patients with advanced ovarian cancer:

  1. The Introduction should be shortened by at least 30% and the various statements should be linked together more organically. This would allow focusing more on the purpose of the review and going straight to the study findings.

Thanks for the suggestion. Done.

  1. Regarding CT imaging, I suggest mentioning the added value of dual energy CT over conventional CT for the detection and characterization of peritoneal deposits in patients with advanced OC.

Thanks for the suggestion. Done.

Reviewer 3 Report

Comments and Suggestions for Authors

This is a good review on the state of the art of imaging of peritoneal carcinosis in advanced ovarian cancer. The contents can have a non-negligible educational value for clinicians working in the field.

In my opinion, however, the review lacks an in-depth presentation of Authors' opinions and views on the evolution of the field. Can the Authors expand on this?

Some further comments:

- the text is quite fragmented. Can the Authors work on style to make it flow better?

- English editing is necessary

- Careful proof-reading is advised (see for instance line 108, blank space between parentheses)

Comments on the Quality of English Language

See above

Author Response

This is a good review on the state of the art of imaging of peritoneal carcinosis in advanced ovarian cancer. The contents can have a non-negligible educational value for clinicians working in the field.

In my opinion, however, the review lacks an in-depth presentation of Authors' opinions and views on the evolution of the field. Can the Authors expand on this?

Thanks for the suggestions. Done.

Some further comments:

- the text is quite fragmented. Can the Authors work on style to make it flow better?

Thanks for the suggestions. Done.

- English editing is necessary

Thanks for the suggestions. Done.

- Careful proof-reading is advised (see for instance line 108, blank space between parentheses)

Thanks for the suggestions. Done.

Round 2

Reviewer 1 Report

Comments and Suggestions for Authors

The manuscript substantially was improved. 

Author Response

1: Line 103: Multidetector Computed Tomography (MDCT).. 

Thanks for the suggestion. Done.

2: -Line 111: Please expand  the term ‘18FDG PET/CT’.

Thanks for the suggestion. Done.

3: Line 143: ....in this review (Figure 1). Please add References, including the papers used for this review.

Thanks for the suggestion. Done.

4: Did you inlude 78 papers in this review (as stated in the Text) or 87 articles (as stated in Figure 1)?

Thanks for the suggestion. In this review we have included 87 articles.

5: 159: reported similar...

Thanks for the suggestion. Done.

6: Lines 435, 439, 446:....paracolic gutter (s)

Thanks for the suggestion. We changed “parietocolic” in” paracolic ”.
